# Lomasomes and Other Fungal Plasma Membrane Macroinvaginations Have a Tubular and Lamellar Genesis

**DOI:** 10.3390/jof8121316

**Published:** 2022-12-19

**Authors:** Igor S. Mazheika, Nadezhda V. Psurtseva, Olga V. Kamzolkina

**Affiliations:** 1Department of mycology and algology, Lomonosov Moscow State University, 119991 Moscow, Russia; 2Vavilov Institute of General Genetics, Russian Academy of Sciences, 117971 Moscow, Russia; 3Komarov Botanical Institute, Russian Academy of Sciences, 197376 St. Petersburg, Russia

**Keywords:** wood-decaying fungi, endocytosis, Z-stacks, time-lapse video

## Abstract

The plasma membrane of filamentous fungi forms large-sized invaginations, which are either tubes or parietal vesicles. Vesicular macroinvaginations at the ultrastructural level correspond to classical lomasomes. There is an assumption that vesicular macroinvaginations/lomasomes may be involved in macrovesicular endocytosis. The original aim of this study was to test for the presence of macroendocytosis in xylotrophic basidiomycetes using time-lapse and Z-stacks fluorescent microscopic technologies. However, the results were unexpected since most of the membrane structures labeled by the endocytic tracer (FM4-64 analog) are various types of plasma membrane macroinvaginations and not any endomembranes. All of these macroinvaginations have a tubular or lamellar genesis. Moreover, under specific conditions of a microscopic preparation, the diameter of the tubes forming the macroinvaginations increases with the time of the sample observation. In addition, the morphology and successive formation of the macroinvaginations mimic the endocytic pathway; these invaginations can easily be mistaken for endocytic vesicles, endosomes, and vacuole-lysosomes. The paper analyzes the various macroinvagination types, suggests their biological functions, and discusses some features of fungal endocytosis. This study is a next step toward understanding complex fungal physiology and is a presentation of a new intracellular tubular system in wood-decaying fungi.

## 1. Introduction

The plasma membrane (PM) of fungal cells is capable of forming large-sized invaginations exceeding 100 nm in diameter or length [1]. There are three main types of such invaginations: macrovesicular invaginations, tubes, and eisosomes. Eisosomes are described at the fluorescent and electron microscopic levels in yeast (for shot review, see [2]). There are also data for some filamentous fungi, including lichenized [3,4,5]. Eisosomes are furrow invagination of PM. The depth of eisosomes is usually small, within 50 nm (varies depending on osmotic pressure), but the length along the surface of the PM reaches 0.4 µm in budding yeast and 2 µm in fission yeast. Eisosomes have a rigid scaffold, including proteins specific for eisosomes. Currently, the term “eisosome” is often used to refer to this protein complex and not to the membrane invagination itself [6].

Tubular invaginations (up to 5 µm long) at the fluorescent-microscopic level are described for filamentous basidiomycetes in our previous works [1,7]. At the electron-microscopic level, invaginating tubes are often presented as curved towards their base and are described not only in filamentous basidiomycetes but also in yeasts [8,9,10].

Macrovesicular or spherical macroinvaginations of the PM, 0.2–2 μm in size, can be divided into three types depending on their ultrastructure [1]. Type I macroinvaginations at the ultrastructural level are large vesicles with a hollow lumen. Presumably, such macroinvaginations are artifactual or formed in old/dying cells. Macroinvaginations II, in fact, are the same tubes or lamellae, bent to their base (at the fluorescent level can be perceived as spherical invaginations). Macroinvaginations III correspond at the electron-microscopic level to classical lomasomes (see below). Assumedly there is a relation between macroinvaginations II and III—it is possible that macroinvagination II, forming secondary invaginations, can form macroinvagination type III [1]. It should be noted that in our previous work [1], the term “macroinvagination” was used primarily to refer to the macrovesicular invaginations of the PM. In this study, the term is extended to all large-sized PM invaginations.

Classical lomasomes (for terminology, see [1]) have been found in fungi for a long time. One of the first observations is found in the study by Girbardt (1958) on *Polyporus versicolor* [11]. Most often, lomasomes are found in electron microscopic samples in basidial xylotrophs, but they are also found in other fungi and fungi-like organisms [12,13,14,15]. The ultrastructural morphology of lomasomes is very diverse; they can consist of vesicles, tubes, lamellae, and their various combinations. The functions and mechanism of the formation of lomasomes are not known and represent one of the enigmas of mycology [14]. Presumably, lomasomes can participate in endocytosis separate from the PM and form endocytic macrovesicles. Such endocytosis is called macrovesicular endocytosis, but its existence is questionable [1].

Styryl fluorophores such as FM4-64 and its analogs are traditionally used to trace the endocytic pathway in fungi. Budding yeast was one of the first objects of such tracking [16]. Styryl fluorophores were then used to study endocytosis in filamentous fungi [17,18]. Endocytotic uptake of FM4-64 is similar in yeast and filamentous fungi and involves several major steps. For example, in the study by Peñalva (2005), first, the fluorophore marks the PM and cortical punctuate structures, then (after 5–10 min) the label also appears in small vesicles, after a 15-min interval, vacuoles begin to be labeled [18]. These steps correspond to the stage of formation of endocytic patches (primary vesicles covered in a rigid actin scaffold) and early endosomes, the stage of labeling late endosomes/early vacuoles, and the stage of labeling tonoplasts of vacuoles-lysosomes. Similar results were obtained in our previous studies for *Rhizoctonia solani* and some species of wood-decaying basidiomycetes [7,19,20]. We have proposed two important things related to fungal endocytosis and fungal cell organization. First, based on a number of features of early endocytosis, including the independence of the formation of primary PM invaginations from the actin cytoskeleton, the curtain model of the regulation of PM tension in fungi has been proposed [7]. Second, two types of fungal endocytosis have been identified [20]. Classical or patch-type endocytosis is characteristic, for example, of *R. solani*. Early endocytosis here occurs mainly through endocytic patches, i.e., endocytic pits do not exceed 100 nm in size. Macrovesicular endocytosis has been found in basidial xylotrophs and *Coprinus comatus*. Here, the size range of primary endocytic structures is wider and reaches 2 μm (however, as mentioned above, the scission of large PM invaginations has not been proven).

The original goal of this study was to seek evidence for the involvement of lomasomes and other fungal PM macroinvaginations in endocytosis. Static photographs taken in a single focal plane contain limited information and do not allow a full study of the dynamics of the formation of PM invaginations, their mobility, connection with the PM, and so on. Here, methods for obtaining time-lapse images, creating videos based on them, as well as Z-stack technology, which allows amplifying the fluorescent signal through stack summation, and creating 3D reconstructions of fungal hyphae, come to the rescue. The use of these methods led to an unexpected complete revision of views on the uptake of styryl labels by fungi, on the vesicular–vacuolar system, and on endocytosis in general. Thus, the goal of the study has expanded. The goal is to classify and establish the features of various types of PM macroinvaginations in xylotrophic basidiomycetes at the volumetric fluorescent and electron microscopic levels and identification of relations between macroinvaginations and endocytosis.

## 2. Material and Methods

### 2.1. Strains and Culture Conditions

Main object of the research: xylotrophic basidiomycete *Stereum hirsutum* (Willd.) Pers. [20]. Part of the microscopic studies is duplicated on a bioluminescent xylotroph *Omphalotus olearius* (DC.) Singer and Appendix A features *Neonothopanus nambi* (Speg.) R.H. Petersen and Krisai (LE-BIN 2082 and LE-BIN 3297, respectively, Komarov Botanical Institute Basidiomycetes Culture Collection, St. Petersburg, Russia). Other wood-decaying basidiomycetes *Lentinula edodes* (Berk.) Pegler, *Panellus luminescens* (Corner) Corner [21] and *P. stipticus* (Bull.) P. Karst (LE-BIN 4431, Komarov Botanical Institute) were used for electron microscopy.

Fungal cultures were stored for a long time on malt agar medium with 1.5% agar (Panreac, Spain) at 4 °C (except for *N. nambi*—at room temperature). Working deponents and mycelium for experiments were grown on Czapek medium (CzM), рН 7, with 1.5% agar at 22–25 °C in the dark. For separate experiments, liquid CzM was used, both pH 7 and pH 5 (acidified CzM with 2-morpholinoethanesulfonic acid, Sigma-Aldrich, St. Louis, MO, USA). Samples were grown on 9 cm Petri dishes, agarized CzM was covered with cellophane discs, and the inoculum (an agar block about 10 by 10 mm in size) was placed near the rim of the dish. The experiments used colonies aged from 4.5 to 7.5 days of growth for *S. hirsutum* and from 9.5 to 16 days for *O. olearius*.

### 2.2. Preparation of Microscopic Samples, Obtaining Photos and Videos

In experiments with preincubation of mycelium in liquid CzM, the cellophane disk with mycelium was transferred from the cultivation dish to a new dish, and 15 mL of CzM was added (the liquid should cover the mycelium but with minimal damage/detachment of the mycelium from the cellophane). The samples were incubated in the dark at room temperature. In both cases, with and without preincubation, a fragment of cellophane with mycelium, including the marginal zone of primary growth (usually 5–10 mm from the edge of the colony) and a narrow strip of mycelium with secondary growth, was cut with a blade and transferred to a glass slide. The sample was quickly mounted with 30 µL of one of the solution variants, not allowing the cellophane with mycelium to dry, and covered with a 24 by 24 mm cover slip. The variants of mounting solutions were: distilled water; CzM, pH 7; CzM, pH 5; CzM, pH 7, with 0.4 M sorbitol (Sigma-Aldrich); CzM, pH 7, with 25 µM Latrunculin A (LatA; Santa Cruz Biotechnology, Dallas, Texas, USA). The variant without preincubation with mounting with CzM pH 7 was taken as the basic variant of the sample treatment. In long-term microscope experiments, the edges of the coverslip were sealed with rubber glue, and the specimen was kept outside the microscope in a humid chamber. Immediately before preparing the slide, 0.5 μL of fluorophore (or a combination of fluorophores) was added to the mounting solution. Final concentrations of fluorophores: AM4-64 (Biotium, Fremont, California, USA)—100 µM; CFDA (6-carboxyfluorescein diacetate, Sigma-Aldrich)—2 µM; Nile Red (Sigma-Aldrich)—10 µM.

For microscopy, a motorized microscope Imager M1 (Zeiss, Germany) with a camera Hamamatsu 1394 ORCA-ERA (Japan) and immersion objective 63× was used. A constant exposure for all samples (500 ms most often) was used. Single photographs, as well as time-lapse and Z-stack photographs, were obtained using the program AxioVision LE 4.8 (Zeiss). The distance between focal slices in Z-stack snaping was set to 70 nm, and the time-lapse snaping interval was 1 sec. All photographs were divided into groups according to three observation intervals: 5–15 min, 15–30 min, and 30–45 min. For quantitative analysis, 20 to 50 photographs were obtained in a single focal plane for each time interval of observation.

Photo processing, editing and processing of time-lapse video, and summation and 3D reconstruction of Z-stacks were performed in the ImageJ 1.53n (NIH, Bethesda, Maryland, USA) using built-in functions. Separate Z-stack videos were processed with the DeconvolutionLab plugin (http://bigwww.epfl.ch/, accessed date 1 October 2022) using the Tikhonov–Miller deconvolution algorithm. Photoshop 8 (Adobe Systems, San Jose, CA, USA) for image editing was also used. Figure 10 was created using Sculptris Alpha 6 software (Pixologic, Los Angeles, CA, USA).

The preparation of electron microscopic samples was carried out according to the previously described methods [20,22]. Some electron microscopic samples were prepared from mycelium on cellophane treated with AM4-64 and taken from fluorescent microscopic slides.

### 2.3. Statistical Analysis

Primary data processing was carried out in the program Excel (Microsoft, Redmond, Washington, DC, USA), and diagramming was performed in OriginPro 7.0 (OriginLab, Northampton, MA, USA). Statistica 6 software (StatSoft, Tulsa, OK, USA) was used for statistical data processing. As a control (the variant with which other treatment options were compared), the basic variant of the sample treatment was used. For statistical comparison of the total number (without breakdown by observation intervals) of PM invaginations in one of the sample preparation variants with the basic variant, the nonparametric Whitney–Mann test (U-test) was used. For statistical confirmation of the presence or absence of the effect of microscopy time on the number of one or another type of invagination, a nonparametric test of variance with dependent samples was used—the Friedman test. To establish a relationship between the formation of some types of PM invaginations and the time of observation or between different types of invaginations, the Spearman nonparametric correlation coefficient (Spearman R) was used. In all cases, differences between samples or the reliability of Spearman R were considered statistically significant at *p* < 0.05.

## 3. Results

### 3.1. Variants of AM4-64 Accumulation in Fungal Hyphae

When labeling the mycelium of xylotrophic fungi by AM4-64 in microscopic samples, three types of hyphae can be distinguished by the distribution of the fluorescent label in cells. In labeled hyphae of the first type, the main AM4-64 signal is concentrated in the PM and invaginations of the PM. The cytoplasm does not fluoresce at working exposures or fluoresces weakly (Figure 1A,B). In the second type of hyphae, the AM4-64 signal is present not only in the PM but is also diffusely distributed in the cell cytoplasm. In such cells, large vacuoles can usually be visualized. They are either visible as darker areas in the diffusely luminous cytoplasm, or their tonoplasts also begin to accumulate a signal from the cytoplasm and fluoresce slightly (Figure 1C–E). The third type—let us call them motley hyphae—contain a fluorescent signal in the cells, but unlike diffusely labeled hyphae, it is bright and unevenly distributed along the length of the cells (Figure 1F,G). This motley may be due to two causes. The hypha can be filled with large vacuoles touching the PM; the tonoplasts of the vacuoles, which receive a label from the PM, and the cytoplasm compressed between the vacuoles are brightly fluorescing. Another variant, the cell protoplast, is split as a result of plasmolysis; the cytoplasm between the voids shrinks and brightly fluoresces. Appendix A clearly shows how the AM4-64 label directly passes from the PM to the vacuolar tonoplast or splitting protoplast membrane. Fine mycelium (less than two microns in diameter [20]) occupies a special place among the motley hyphae: its cytoplasm can fluoresce brightly, while the signal is not diffuse but not distributed in separate areas.

In microscopic samples, the proportion of diffusely labeled and motley hyphae increases with the time of observation (see Appendix A, observation during 20 min). When the slide with mycelium is mounted with water (hypotonic conditions), after about half an hour of observation, most hyphae become diffusely labeled or motley. Accordingly, only hyphae of the first type, with an absent or weak AM4-64 signal in the cytoplasm, were used in this study for the analysis of PM invaginations.

### 3.2. Main Types of PM Macroinvaginations in Xylotrophic Basidiomycetes

AM4-64-labeled PM invaginations in *S. hirsutum* and other xylotrophs at the fluorescent-microscopic level can be with or without a lumen. The following types belong to invaginations without a lumen: glomeruli, pendants, and plaques. The names of the types of invaginations are given in this work for convenient differentiation of the invaginations among themselves and have no relation with any terms from the mycological literature. The invaginations with a lumen include two groups of structures: small vesicles and thin tubes, and large vesicles and thick tubes. Detailed characteristics of all types of PM invaginations are outlined below.

#### 3.2.1. Glomeruli, Pendants, and Plaques

Glomeruli at the fluorescent-microscopic level in a single focal plane are small dots or globules of the AM4-64 signal without a lumen and pressed against the PM (Figure 2A, Appendix A). In some cases, the glomeruli can be quite large (Figure 2B). On the 3D reconstruction of Z-stacks, it can be seen that some glomeruli are not spherical—they are thin rollers pressed against the PM located across the cell (Appendix A).

In addition to globular and in the form of short rollers glomeruli, pseudo-glomeruli are often found. Pseudo-glomeruli can be transverse thin tubes with a narrow or tortuous lumen, and then they are visible as true glomeruli even in multiple focal sections (Appendix A). In addition, pseudo-glomeruli can simply be transverse tubes that do not fall into the focal section (in the case of working with photographs with a single focal plane).

The pendants are similar in everything to the glomeruli, but they are not pressed to the PM, although they remain attached to it (Figure 2C–E). The connection with the PM is not always distinguishable at the fluorescent-microscopic level, but the video clearly shows that the pendants are constantly moving randomly, remaining within a certain radius relative to a certain point on the PM (Appendix A). As with glomeruli, many pendants are actually pseudo-pendants. They are either tubes with narrow lumens, which are difficult to see with fluorescent labeling (Appendix A), or movable tubes that are out of the focal plane (Appendix A). In some cases, the pendant can be scissored from the PM and move along the hypha by the cytoplasmic flow (Appendix A). Judging by the movement of such scissored invagination, it is not controlled by the cytoskeleton but moves freely in the depth of the cytoplasm.

On a single focal slice, the plaques are small areas of the PM with an AM4-64 signal that is stronger than the surrounding PM. Plaques often look like flattened fluorescent structures pressed against the PM (Figure 2A,F; Appendix A). On 3D reconstruction, plaques, like glomeruli, can be stretched across the cell (Appendix A). In isolated cases, it can be observed how plaques form from parietal vesicles/tubes that flatten and lose their lumen (Appendix A). There is a video (Appendix A) that, on the contrary, shows how a large invagination is formed from a plaque.

Quantitative analysis shows that glomeruli and pendants are the most represented among all types of PM invaginations. In the basic variant of the sample treatment, the average number of glomeruli and pendants in *S. hirsutum*, recalculated per 1000 septa, is an order of magnitude higher than the average number of small vesicles that rank second in terms of representation (*U*-test, *p* << 0.01; Figure 3A).

In all sample treatment options, the number of glomeruli and pendants does not statistically depend on the observation time (Friedman test, *p* > 0.05). However, Figure 3B shows that in the treatment variant with preincubation and with sorbitol mounting, in some experiments, there is a significant increase in the number of glomeruli/pendants after 15 min from the start of observation. Glomeruli and pendants appear either immediately after the microscopic slide preparation with the sample (or they are in the cells before the preparation of the sample) or a few minutes after the preparation. On what it depends is not established. Therefore, in quantitative experiments, the first observation interval is 5–15 min and not 0–15 min.

Among the various treatment options, compared with the basic variant of treatment, preincubation of mycelia, as well as slide mounting with hypertonic medium and medium with LatA, lead to a statistically significant increase in the number of glomeruli and pendants (*U*-test, *p* << 0.01). The greatest influence on the formation of glomeruli and pendants is exerted by mounting the preparation with medium with sorbitol after preincubation. In this variant, the number of invaginations can reach about 40,000 per 1000 septa. On average, the hypertonic conditions used increase the number of glomeruli and pendants by about five times relative to the basic conditions. Hourly preincubation in CzM pH 7 before slide mounting in CzM pH 7 increases the formation of glomeruli and pendants by almost three times. The same result is obtained when analyzing the number of invaginations in *O. olearius*. However, for *O. olearius*, additional experiments were performed with a 12-h preincubation of mycelia in CzM pH 7 (Figure 4A). Long-term preincubation leads to a reverse decrease in the number of glomeruli and pendants approximately to the basic level (*U*-test, *p* > 0.05).

On the contrary, mounting with acidified ChM and hypotonic medium did not affect the total number of glomeruli and pendants (*U*-test, *p* > 0.05). However, Figure 3B shows that the hypotonic medium (mounting the preparations with water), although it does not have a statistically significant effect, increases the scatter of data—in some samples, the number of glomeruli and pendants increases to 20–25,000 per 1000 septa, with a maximum value only of about 8000 in the basic variant.

The occurrence of plaques in fungal cells is not high (Figure 4B,C). In the basic variant of the sample preparation, it reaches a maximum of 250 plaques per 1000 septa in *S. hirsutum*. The influence of the different variants of treatment is as follows: hypertonicity, acidification of the medium, and LatA do not affect the formation of plaques (*U*-test, *p* > 0.05). Two factors influence: mycelium preincubation in CzM and hypotonicity (*U*-test, *p* << 0.01). Thus, an hourly preincubation of *S. hirsutum* mycelium in CzM leads to an increase in the number of plaques by an average of 2.5 times relative to the basic variant, and a 12-h preincubation of *O. olearius* gives an increase of almost 7.5 times. The mounting of the *S. hirsutum* slides in water enhances the formation of plaques by an average of more than 3.5 times relative to the basic variant. However, if the mycelium is preincubated in a liquid medium before being put into water, the spread of values increases; in different samples, the number of plaques is calculated from 15 to 1000 plaques per 1000 septa.

#### 3.2.2. Small Vesicles and Thin Tubes

Small vesicles can be both parietal in single focal sections and located in the depth of the cytoplasm (Figure 5). The diameter of small vesicles does not exceed 1–1.5 microns. Most of the small vesicles are transverse focal sections through transverse thin tubes, both pressed against the PM (Appendix A) and extending into the depth of the cytoplasm, but curved in their upper part across the cell (Appendix A). Apparently, there are other ways to form small vesicles (see the section on large vesicles). In Appendix A, it can be seen how the small vesicles disappear, flattening, pressing against the PM, and turning into a plaque.

In addition to thin tubes located entirely or partially across the hypha and visualized most often as small vesicles, fungal cells contain other types of thin tubes that are visible in single focal planes as tubes with a diameter less than 500 nm, not vesicles. The occurrence of such tubes is, on average, 6.5 times lower than that of small vesicles (in the basic variant of treatment). This includes straight thin tubes located perpendicular to the PM—standing tubes (Figure 5). Appendix A shows the growth of a standing thin tube. Another type of thin tube is those that run along the fungus cell—longitudinal tubes (see the growth of these tubes in Appendix A). Such tubes can be pressed against the PM or go in the depth of the cytoplasm, be relatively straight or tortuous or curving (Figure 5; Appendix A). The length of longitudinal tubes can reach 15–20 microns.

Small vesicles are the second most common type of invaginations after glomeruli and pendants. Their average number may exceed 3000 per 1000 septa in some experiments with *S. hirsutum*. Moreover, these results are most likely underestimated since, as already mentioned, in single focal sections, many small vesicles are visible as glomeruli. So, for example, in the basic treatment variant, the Spearman correlation coefficient between glomeruli/pendants and small vesicles is 0.82 (*p* < 0.01), which further confirms their affinity. Another evidence of the relationship between glomeruli and small vesicles is that the pattern of the influence of different experimental conditions on the formation of glomeruli and small vesicles is similar (Figure 3B and Figure 6A). Thus, in the basic variant, as for glomeruli, the Friedman test shows the independence of the formation of small vesicles from the time of sample observation (*p* > 0.05). The formation of small vesicles and thin tubes is also not affected by hypophase and acidification of the medium (*U*-test, *p* > 0.05). As in the case of glomeruli and pendants, prolonged preincubation of mycelium in CzM leads to a reverse decrease in the number of small vesicles and thin tubes (Figure 6B). On the contrary, preincubation of mycelium in CzM, mounting the slides with medium with sorbitol or LatA lead to an increase in the number of small vesicles and thin tubes (*U*-test, *p* < 0.05). However, if the preincubation and hyperphase, on average, enhance the formation of small vesicles and tubes by 2.5–3.5 times (against 3–5 times in glomeruli and pendants), then LatA has a lesser effect here; it increases the number of vesicles and tubes by about 1.5 times, while the number of glomeruli and pendants more than 3 times. If to analyze the effect of LatA on the formation of small vesicles and thin tubes separately, it turns out that the inhibitor only affects the formation of small vesicles (*U*-test, *p* < 0.05) but not on thin tubes (*U*-test, *p* > 0 0.05). It should be noted that both sorbitol and LatA increase the average size of small vesicles (visual observation).

Small vesicles (and hence transverse tubes) appear early in the microscopic slides, together with or a little later than glomeruli and pendants. However, longitudinal tubes appear later in the samples (not earlier than 10–15 min after the slide preparation), and their occurrence increases along with the observation time (Friedman test, *p* < 0.01; Spearman *R* = 0.61, *p* < 0.01).

#### 3.2.3. Large Vesicles and Thick Tubes

Large vesicles and thick tubes are in many ways similar to small vesicles and thin tubes, but they are larger; the diameter of large vesicles is from 2–2.5 microns or more (Figure 7A–C; Appendix A), the diameter of thick tubes is from one micron or more (Figure 7C; Appendix A). There is also a close relationship between vesicles and tubes; many large vesicles are short, transverse, thick tubes in a single focal section. Appendix A shows the turning of thick longitudinal tubes across the cell; they turn into large vesicles in a single focal section. In Appendix A, it can be observed another mechanism for the formation of large vesicles: a short standing tube swells into a large vesicle or a part of a longitudinal tube pressed against the PM swells into a vesicle. In other words, even if a vesicle is truly spherical and it is not an optical section through a transverse tube, it still goes through a tube stage during its formation.

However, there is a difference between large vesicles/thick tubes and small vesicles/thin tubes in addition to size. First, thick tubes and large vesicles are less mobile. If the free end of a long thin tube can and often moves chaotically in the depth of the cytoplasm, like a pendant (Appendix A), then thick tubes are usually static (Appendix A). Second, the occurrence of large vesicles is lower than the occurrence of small vesicles (Figure 3A). In the basic variant of mycelia treatment, the number of large vesicles is, on average, almost six times less than that of small ones. Third, for small vesicles and thin tubes, only the appearance of thin tubes depends on the time of observation, but not small vesicles (see above). Large vesicles and thick tubes appear in microscope slides later than all other PM invaginations. In the first 20 min of observation, large vesicles are extremely rare, and thick tubes are almost never seen. Friedman test for these invaginations in the basic treatment variant, *p* < 0.05; Spearman R = 0.55–0.56, *p* < 0.05 (the not high value of R is due to the abrupt appearance of the invaginations after about 20 min of observation). Fourth, as with small vesicles and thin tubes, preincubation in CzM and sorbitol mounting enhance the formation of large vesicles and thick tubes (Figure 7D; *U*-test, *p* < 0.05). However, treatment of the mycelium with LatA, which affects the formation of small vesicles, does not statistically affect the formation of large vesicles and thick tubes (*U*-test, *p* > 0.05). Mounting of slides with water does not lead to statistically significant differences from the basic treatment variant (*U*-test, *p* > 0.05). However, Figure 7D shows that the spread of values in the hypophase is significantly reduced; therefore, on average, the number of large vesicles and thick tubes is almost 3.5 times less than in the basic variant.

It should be noted that large vesicles and thick tubes occur unevenly in the mycelium; they are usually concentrated in individual cells, in contrast to the glomeruli, which are more (but not completely) evenly distributed between cells.

### 3.3. Vacuole-Lysosome Candidates, Co-Labeling of the Vacuolar System with PM Macroinvaginations, Nile Red Labeling

In the hyphae of the first type (see the first section of the Results), with an absent or weak diffuse AM4–64 signal in the cytoplasm, AM4-64 marks not only the PM macroinvaginations. Sometimes there are vacuole-like structures that differ from large vesicles and thick tubes—vacuole-lysosome candidates (VLCs). Macroinvaginations are always characterized by a powerful signal, the same as that of the PM or more powerful. The membranes of the VLC fluoresce more weakly than macroinvaginations; in some cases, they are visible only when the signal is amplified, for example, through Z-stack summation. VLCs can be large and touch the PM of opposite cell walls (Appendix A, red arrow) or be smaller (Figure 8A, arrow). The occurrence of VLCs is low; for example, in experiments with vacuole and large vesicles co-labeling (see below), 9 VLCs were found for 119 large vesicles. In the experiment with the basic variant of treatment of *S. hirsutum* mycelium, but keeping the slides sealed with rubber glue for 2.5 h, the amount of VLC differed 5 times in the basal and frontal parts of the mycelium (the cover glass was conditionally divided in half, perpendicular to the main growth of hyphae, and VLCs were counted separately in each half).

The best result when labeling vacuoles with CFDA is obtained by acidifying the medium mounting the slides. At neutral pH, there are fewer vacuoles, and tubular vacuoles are less often marked. Acidification of the medium weakens the signal from AM4-64 but does not affect the dynamics of macroinvaginations (see above). CFDA accumulates in fungal vacuoles in the first 15–20 min after sample preparation, and then the label begins to diffusely accumulate in the cytoplasm of cells. In some cells, CFDA, like AM4-64, immediately begins to label the entire volume of the cytoplasm. In such cells, vacuoles are not identifiable. CFDA marks in *S. hirsutum* hyphae large (greater than 2 µm in diameter) rounded or oval vacuoles (Figure 8B; Appendix A), small (less than 2 µm) rounded vacuoles (Figure 8C; Appendix A), and tubular vacuoles (Figure 8C; Appendix A). Tubular vacuoles most often are necks between large rounded vacuoles or are brunches of the latter. Appendix A show that with the help of tubular vacuoles, large vacuoles flow from one place in the cell to another or transfer their contents to each other. Tubular vacuoles can be either relatively amorphous, elongated ordinary vacuoles (Appendix A), or be in the form of thin long tubes (Figure 8C, arrow), similar to the super-elongated mitochondria of xylotrophs [20].

To analyze the colocalization of the AM4-64 and CFDA labels, separate experiments were performed with *S. hirsutum* mycelium. In three biological replicates with the basic treatment variant in the observation intervals from 5 to 45 min was found 287 septa, 1235 small vacuoles, 919 large vacuoles, 30 tubular vacuoles, 2757 glomeruli and pendants, 109 small vesicles and thin tubes, and 9 large vesicles and thick tubes. A merged signal was found only in four small areas, and only in one of them a small vacuole was involved; in other cases, labeled structures were not identified. The results of co-labeling of small and large vacuoles with glomeruli, pendants, and small vesicles are shown in Appendix A.

Since differentiation is primarily required between vacuoles and large vesicles/thick tubes, additional experiments were performed. CFDA was introduced into slides 30 min after the preparation of samples with AM4-64, and no random fields of view were photographed, but areas of mycelium with large vesicles or thick tubes were searched for in a directed manner. As a result of two experiments, 118 visual fields were analyzed, which revealed 9 VLCs, 119 large vesicles, 28 thick tubes, 538 large vacuoles, 193 small vacuoles, and 7 tubular vacuoles. Only two colocalizations of AM4-64 and CFDA labels (AM4-64 in the membrane, CFDA in the lumen) were found for VLC (Figure 8E,F), and only one colocalization for a large vesicle (Figure 8G,H). Figure 8I–N and Appendix A show a typical distribution of CFDA and AM4-64 labels, with no signal overlap between vacuoles and large vesicles/thick tubes.

Unfortunately, Nile Red, which labels predominantly lipid droplets in fungi, fluoresces in both the red and green channels. In view of this, colocalization in a static image of Nile Red and AM4-64 is difficult. However, analysis of dynamic images shows that structures with a Nile Red signal (lipid drops) behave differently and differ morphologically from AM4-64 structures (glomeruli and pendants). The glomeruli are relatively static and pressed against the PM, while the pendants move randomly around one point on the PM. Lipid droplets are more mobile than glomeruli, usually not pressed against the PM, and can move along the hyphae (Appendix A). It is most likely that AM4-64 marks some lipid droplets, and when considering static photographs, they are counted together with glomeruli and pendants. However, their contribution to the total number of calculated invaginations is presumably not large.

### 3.4. Electron Microscopy of PM Macroinvaginations

For electron microscopy, both samples of *S. hirsutum* treated with AM4-64 in fluorescent microscopic slides and ordinary samples of different species of xylotrophic basidiomycetes were used. No differences were found between AM4-64 treated and untreated samples.

Different types of AM4-64 labeled PM macroinvaginations were matched with different types of lomasomes in ultrathin sections. The glomeruli correspond to the classic vesicular lomasomes, most often consisting of a common membrane and thin tubules intertwined or orderly laid inside (Figure 9A–D). Glomeruli elongated across the cell close to the PM correspond to lamellar lomasomes—either densely rolled PM plications (myelin-like bodies) or consisting of more loosely laid lamellae (Figure 9A,E–G). Lomosomes often combine both tubular and lamellar structures (Figure 9A,C). The pendants correspond to lomasomes that are separated from the PM and connected to it by a thin tubule or a lamella (Figure 9E). In Figure 9G,H, a small vesicle (transverse tube) and a thin longitudinal tube can be seen. However, differentiation of macroinvaginations with lumen from vacuoles at the electron microscopic level is difficult. Figure 9I shows a possible plaque—a flattened vesicular lomasome extended along the PM.

## 4. Discussion

### 4.1. Fungal Macroinvaginations of the PM Are Tubular and Lamellar in Nature

As a result of the study, it was found that cells of xylotrophic basidiomycetes actively and abundantly form large PM invaginations. Presumably, filamentous fungi from other taxonomic and ecological-trophic groups also have this property. This issue requires additional research. Judging by the fact that preincubation of the mycelium in a liquid medium enhances the formation of invaginations, as well as by the initial lag of the formation of invaginations in some cases when the mycelium is transferred to the slides from a solid medium, it can be assumed that the PM of the mycelium immersed in the liquid invaginates most actively. However, it is difficult to say how mycelium growing in artificial and natural substrates, with a small amount of free liquid or with a separation of water and air phases, behaves in this respect in the depth of an agar medium or on its surface, in soil, in a woody substrate, etc.

Undoubtedly, one of the two main conclusions of this study is that large PM invaginations in xylotrophs have a tubular or lamellar nature, and the thickness (lumen diameter) of the tubes/lamellae can be different, and the type of macroinvagination will depend on this. The mechanism for the formation of thin and thick tubes is obvious and is shown in the videos in this article. The formation of small or large vesicles occurs in two main ways. In the first way, vesicles are sections through transverse tubes and are visible as vesicles in a single focal section. In the second way, shown in Appendix A, the tubules swell into a true vesicle. Apparently, there are still mechanisms for the formation of PM macroinvaginations of the vesicle type. For example, it can be through the wrapping of the lamella (a plicate of the PM) with the formation of a hollow roller transverse to the cell (perhaps Appendix A demonstrates this). In any case, the main idea is that any macroinvagination passes through the stage of a tube (or lamella) during its formation. This idea extends to glomeruli and pendants (maybe also plaques), the mechanism of formation of which is not as obvious as that of tubes and vesicles.

We propose three main mechanisms for the formation of glomeruli and pendants (Figure 10). Here, electron microscopic images of lomasomes are important evidence. Precisely lomasomes, apparently, are visible at the fluorescent microscopic level as glomeruli and pendants. The first mechanism: a thin long filamentous tube is drawn into the cell (no more than 50 nm in cross-sectional diameter, most likely). It curls up into a glomerulus near the PM or at some distance (Figure 10A–C). Accordingly, a true glomerulus or pendant is formed. Similar can be seen in Figure 9B, as well as in a number of classic works (see Figures in [14,23]). The second mechanism: many classic vesicular lomasomes have a common outer membrane—they look like a parietal vacuole with vesicles or tubules inside (Figure 9A, for example). It can be assumed that, in this case, a PM invagination tube is first formed, which swells into a vesicle (or it is a tube pressed to the PM initially). After that, from one or many sites of the membrane of such invagination, secondary invagination tubes begin to form in the lumen of the maternal invagination (Figure 10D–F). They fill the lumen of the maternal macroinvagination chaotically (then vesicles and tubules are visible on the section of the lomasome, Figure 9A top) or with a certain stacking (Figure 9C,D, then, for example, only vesicles are visible on the section or regular alternation of vesicles and short tubules). The third mechanism: not a filamentous tube is drawn into the cell, but a thin lamella (less than 50 nm thick; Figure 10G–H). Most often, such a lamella is located across the cell. This lamella then either folds into a tight roll and forms the classic myelin-like body or folds into a looser lomasome (Figure 9E,G). On ultrathin sections, lamellar lomasomes consist of even concentric tubules (but in reality, they cannot be true tubules—the tubules cannot always be exactly in the plane of the section—they are sliced lamellas, Figure 9F). In some of its parts, the lamella may also begin to swell and form secondary tubules-invaginations. Then a hybrid lomasome is obtained—vesicular-lamellar (Figure 9A,C). Accordingly, such lamellar lomasomes are visible at the fluorescent microscopic level as glomeruli pressed against the PM and elongated across the cell.

Separately, it is needed to consider the issue of plaques. Judging by the video in which other macroinvaginations flatten into plaques; judging by the increase in the number of plaques with a long incubation of the mycelium in a liquid medium paralleling the decrease in the number of other macroinvaginations. In addition, judging by the increase in the number of plaques in the hypophase, in which the tension of the PM increases, plaques are presumably glomeruli and small vesicles everted into the periplasmic space due to an increase in the PM tension and an increase in the hyphae diameter.

In the formation of fungal macroinvaginations of the PM, in addition to their tubular-lamellar nature, there are more regularities. The most important of them: in a microscopic slide, different types of macroinvaginations appear at different times after the preparation of the sample. There are two clear dependencies. First, the thicker the tubes from which the invagination is formed, the later such invagination is found in the microscopic slide. Therefore, the very first to appear in the slides are glomeruli and pendants, which are formed from very thin filamentous tubes and lamellae. Then, simultaneously or a little later, small vesicles appear, most often formed by transverse thin tubes. After 10–15 min, longitudinal thin tubes appear in the preparations. Finally, after 20–30 min, large vesicles and thick tubes can be found in individual hyphae cells. Moreover, the number of macroinvaginations per cell decreases along with an increase in the diameter of the forming tubes. Second, in the mycelium cells in freshly prepared slides, predominantly transverse tubules or lamellar rollers are formed. Longitudinal tubules, even thin ones, appear later.

The successive appearance of different types of macroinvaginations in the samples is due to the specific conditions in the microscopic slides and the peculiarities of the control of the tension of the PM by the fungal cell. The preparation of a microscopic slide is accompanied by a strong mechanical effect on the fungal mycelium, and then mechanical stress continues—the pressure of the cover glass, the tension forces of a thin layer of liquid between the cover glass and slide, the movement of the liquid due to drying. Among other things, due to the drying of the slide, the osmotic pressure in the mounting fluid changes. However, here mechanical stress plays a greater role than osmotic changes; in preparations sealed with glue, evaporation in which is minimal, the dynamics of the formation of macroinvaginations do not change. Conversely, tapping the slides on the cover glass accelerates the formation of thick tubes and large vesicles (these observations of the authors are not presented in the Results). An important result should be added here: mounting the slides with a hyperosmotic medium enhances the formation of all types of macroinvaginations from glomeruli to thick tubes, but partial destruction of F-actin by LatA increases the number of only glomeruli/pendants and small vesicles, without affecting the longitudinal thin tubes, large vesicles, and thick tubes. All this can be explained as follows. In freshly prepared microscopic preparations under the conditions of this study, the control over the tension of the PM is performed mainly by the curtain actin subsystem (for the curtain model, see Introduction and [7]). The fungus, using the system of actin driver cables, creates a certain tension of the PM, at which only tubular invaginations of small diameter are formed (most likely with the participation of BAR proteins). This probably roughly corresponds to the native state of the mycelium growing under stable conditions without significant stress. In this case, true glomeruli and pendants are formed, and lamellar rollers or tubes dominate, located across the cell and responsible for the appearance of elongated or pseudo-glomeruli, as well as small vesicles. The transverse arrangement of the rollers and tubes, apparently, is associated with the peculiarities of the tension of the PM (for example, it is not the same along and across the cell). Then, along with an increase in the time elapsed after the preparation of the slides due to mechanical stress, the curtain actin subsystem begins to reorganize (possibly, there is a partial disassembly of actin cables) and loses control over the tension of the PM. The tension of the PM in the cells decreases and its character changes—it becomes possible to form tubular invaginations of a larger diameter and location, including along the cells. It is not clear how the neck of such thick tubular invaginations is formed—it is unlikely that BAR proteins, dynamin, and other neck-forming proteins can control invagination of such a diameter (greater than 1 µm in the case of thick tubes). Perhaps the thick tubes start out as thin tubes and then expand.

### 4.2. AM4-64 Only Mimics the Endocytic Pathway: What about Classical Endocytosis?

In our previous works [1,20], studying static photographs of AM4-64 labeling taken in a single focal section in xylotrophs, we took glomeruli for primary endocytic invaginations/macrovesicles, pendants and small vesicles for endosomes, and large vesicles for vacuoles-lysosomes. The consistent appearance of the macroinvaginations in microscopic slides: glomeruli—small vesicles—and finally large vesicles—additionally confirmed that we were dealing with endocytic absorption of the styryl label and was in good agreement with the endocytic pathways described in the literature in yeast and some filamentous fungi [16,17,18]. However, the change in methodical approaches in this study completely changed the interpretation of the previous and present results and made it possible to draw the second important conclusion of the study: AM4-64 in xylotrophic basidiomycetes marks mainly PM macroinvaginations of various types, only imitating the endocytic pathway due to the similarity of membrane structures and the non-simultaneity of the appearance of macroinvaginations in microscopic slides. This is confirmed by direct observation of the formation of invaginations using time-lapse video, tracking the connection of tubular invaginations with PM using 3D reconstructions of Z-stacks, and the absence of joint labeling of AM4-64 structures and the vacuolar system, etc. In this case, a natural question arises about classical (microvesicular) endocytosis in xylotrophic fungi. If everything that AM4-64 obviously labels in fungal cells has nothing to do with endocytosis, is there endocytic uptake of AM4-64, does it reach vacuole-lysosome tonoplasts by endocytosis?

The answer to this question requires further research. However, the presence of VLCs in xylotrophic cells indicates that classical endocytosis still occurs, not only in the subapical ring but also in non-apical hyphal cells. On the other hand, it is probably not as intense and fast as we previously thought. If the VLC tonoplast is not contaminated with the label from the PM or the diffuse label from the cytoplasm, then AM4-64 is transferred to it through a chain of primary endocytic vesicles and endosomes, which are difficult to trace against the background of a strong signal from macroinvaginations. Active transport of vesicles and endosomes along the hyphae is not ruled out: it is possible that the marginal hyphae of the colony, which were used for microscopy, transport endomembranes with the AM4-64 label to the basal part of the colony, where they already merge with vacuole-lysosomes.

### 4.3. Does Macrovesicular Endocytosis Exist?

This issue, the existence of macrovesicular endocytosis in fungi, was considered at the predominantly theoretical level in our previous publication [1]. The current study was originally designed to address this issue, too. Although the objectives of the study have changed in the course of its implementation, some experimental results have been obtained that shed light on macrovesicular endocytosis. First, it can be said with certain confidence that thick tubes and large vesicles are not scissored from the PM; they are inactive and remain in a slightly changing form for a long time. Second, several time-lapse videos were received, in which it can be traced the separation of large pendants from their place (Appendix A). Of course, the data are not enough to reliably confirm the scission of macroinvaginations from the PM: the video may show endomembranes that included the AM4-64 label. However, if what is observed is really the scission of the pendants from PM, then it can be assumed that macroinvaginations with a rather thin neck-pedicle and mobile ones (such as pendants or thin tubes) are capable of scissoring from the PM. Third, the observed putative scissoring of the pendants is not similar to the scissoring of microvesicles in classical endocytosis. In classical fungal endocytosis, the actin scaffold-covered primary vesicle is constantly controlled by actin cables. After fusion with the endosome, transport occurs along the microtubules [24]. In this study, it does not appear that the scissored macroinvagination was under cytoskeletal control. Rather, the flows of the cytoplasm spontaneously tear it off and carry it along the hypha. Thus, if macrovesicular endocytosis occurs, it differs from classical endocytosis and is more spontaneous than regular (at least under the conditions of our experiments).

### 4.4. Biological Functions of Fungal PM Macroinvaginations

It is most likely that fungal macroinvaginations of the PM are not an artifact. This also applies to large vesicles and thick tubes, despite the fact that they are presumably the response of fungus to the specific conditions of the microscope slide. Similar conditions can also occur in the natural environment, causing the formation of especially large PM invaginations.

Most of all, macroinvaginations of xylotrophs are similar to yeast eisosomes (especially lamellar invaginations), only they are significantly larger. Yeast eisosomes have been assigned different functions. Currently, the most popular concept is that yeast eisosomes are compartments of complexes of regulatory proteins involved in the regulation of yeast metabolism. Previously, it was assumed that eisosomes could take part in the regulation of PM tension, in particular, take in excess membrane during cell shrinkage. However, this hypothesis is currently not popular [25]. However, it is the hypothesis of a reserve membrane pool and instantaneous packing of the PM membrane excess that is most adequate in the case of basidial xylotrophs. Consider the evidence supporting this hypothesis.

Our previous publication showed that hyphae of *Rhizoctonia solani* under hypertonic conditions could shrink in cross-section with a loss of up to 15% in hyphae diameter [7]. The cell wall of fungi is elastic, and the protoplast is attached to it from the inside through multiple sites of focal adhesion; therefore, when cells lose water, hyphae, like yeast cells, shrink entirely. Plasmolysis usually does not occur; the hyphae retain their shape and functionality. If to consider a fungal cell as a cylinder, then simple calculation shows that when the cylinder diameter decreases by 15%, the area of its lateral surface will also decrease by 15%. For example, the PM area of a fungal cell 100 µm long and 5 µm in diameter will be 1570 µm^2^. When the cell is shrunk by 15% in diameter, the lateral area will be 1334.5 µm^2^, i.e., it will decrease by 235.5 µm^2^ (excluding possible longitudinal compression of the hyphae). To instantly remove the excess membrane of such an area, it is needed, for example, approximately 38 invagination tubes 5 µm long and 400 nm thick. These numbers are in good agreement with the results of this study. In other words, the most likely function of macroinvaginations in xylotrophs, and most likely in other filamentous fungi, is to provide a rapid change in the surface area of the PM without damaging the membranes. Under hypotonic conditions (in water), macroinvaginations are also formed, which indicates the existence of a reserve membrane pool that prevents cell destruction with a sharp increase in hyphae diameter. Figure 3B demonstrates that in some cases, when the mycelium is placed in the hypophase, there may be outliers—a sharp increase in the number of glomeruli and pendants. Perhaps this is due to the delay in the establishment of osmotic equilibrium during the transfer of mycelium to the liquid due to the hydrophobic surface of the aerial mycelium growing in the air or water-air phase on cellophane in a Petri dish.

Fungal macroinvaginations may have other functions. For example, some time-lapse videos show the contact of thin tubes with large vacuoles (not shown). Such contact of the vacuole with the external environment can be an additional regulator of the osmotic pressure on the membranes. Previously, we assumed that lomasomes are actively involved in macrovesicular endocytosis and provide the fungus with intracellular digestion (receptor-independent bulk capture of nutrients by macroinvagination from the external environment and their lysis inside the detached invagination or after fusion with the vacuole-lysosome [20]). This type of nutrition would give the basidiomycete an advantage in the rate of absorption of nutrients over the microorganisms of the hyphosphere. However, according to the present study, macroendocytosis in xylotrophs is irregular and is unlikely to be seriously involved in nutrition. On the other hand, under certain conditions in nature, macroendocytosis can become more active and contribute to nutrition, membrane metabolism, etc.

Thus, in the present study, the veil of one of the oldest mycological enigmas, lomasomes, has been lifted. With the help of fluorescent microscopic analysis, due to the capabilities of modern motorized epifluorescent microscopes, it was shown that basidial xylotrophs form many macroinvaginations of the PM, which are formed from tubes and lamellae of different thicknesses. The thinnest—filamentous tubes (lamellae) are folded into glomeruli and rollers (or otherwise transformed), forming lomasomes. Thicker invaginations do not fold but may kink, swell, etc. The activity of formation of macroinvaginations depends on the tension of the PM, which in turn is regulated by the actin cytoskeleton, and the thickness of the tubes that form macroinvaginations depends on the actin system and its integrity. Most likely, the macroinvaginations perform the function of conserving an excess of PM or a reserve membrane pool during changes in the thickness of hyphae (which probably occurs constantly and dynamically).

A new foreshortening of consideration of the membrane structures of the fungal cell has been established, in which large vesicular formations inside the hyphae represent two major groups: either large PM invaginations or vacuoles. Both groups are little involved in the classical endocytic pathway (only a small proportion of vacuoles are VLCs), but some lomasomes may be involved in specific macrovesicular endocytosis. The new perspective explains the plasticity inherent in fungal mycelium and its ability to quickly adapt to changing osmotic conditions, water-air phases, humidity, and so on. However, it also raises new biological questions: it is necessary to establish the intracellular localization of the membrane components of classical endocytosis in non-apical hyphal cells, functions of macrovesicular endocytosis, etc. An efficient solution to the problems associated with these biological issues requires complex methodological approaches that combine genetic labeling of membrane structures with fluorescent proteins and other labels, cytochemical labeling, and modern methods of light and electron microscopy.

## Figures and Tables

**Figure 1 jof-08-01316-f001:**
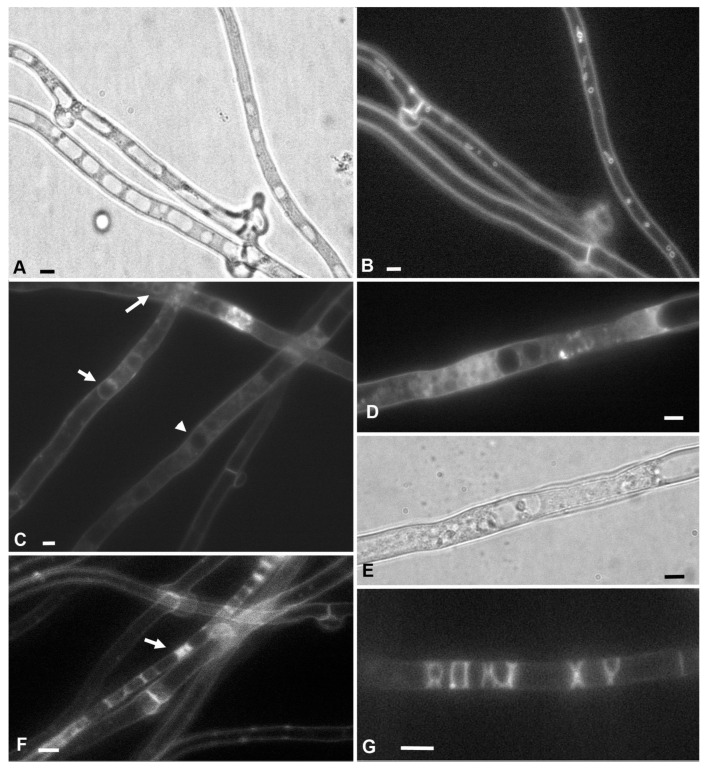
Three types of hyphae in xylotrophic basidiomycetes (in the example of *S. hirsutum*) differ in uptake of the AM4-64 fluorophore. (**A**,**B**)—hyphae of the first type, with an absent or very weak AM4-64 signal in the cytoplasm and endomembranes. PM and its macroinvaginations are labeled. The bright-field photograph (**A**) clearly shows large vacuoles; on the fluorescent image of the same area of the mycelium (**B**), there are no vacuoles, but vesicular and tubular invaginations of the PM are visualized. (**C**–**E**)—hyphae of the second type with a diffuse signal AM4-64 in the cytoplasm. (**C**)—relatively weak fluorescence of the cytoplasm; vacuoles are seen either as dark areas (arrowhead) or have a weakly labeled tonoplast (arrows). (**D**,**E**)—hypha with a more powerful diffuse signal in the cytoplasm (**D**) and the same sample region in the bright field (**E**). (**F**,**G**)—the third type—motley hyphae. (**F**)—arrow points to the motley hypha. (**G**)—with the help of labeled areas of tonoplasts and the edges of plasmolytic protoplasts, the fungus wrote “ROXY XY”. Scale bars are 5 µm.

**Figure 2 jof-08-01316-f002:**
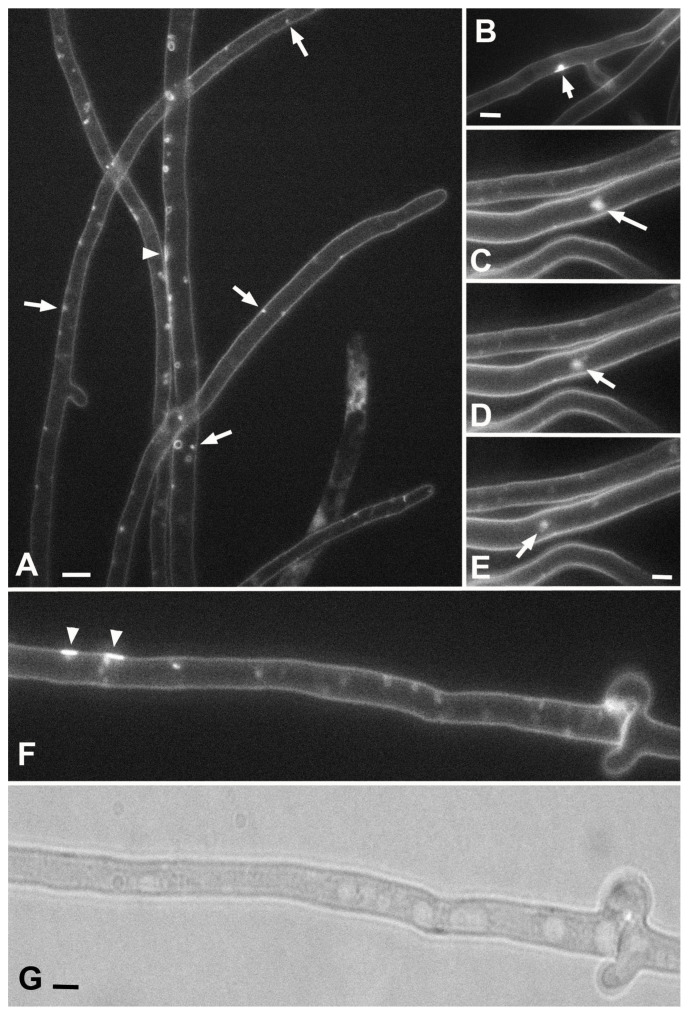
Glomeruli, pendants, and plaques in hyphae of *O. olearius* (**A**) and *S. hirsutum* (**B**–**G**). (**A**)—arrows point to individual glomeruli, and the arrowhead points to plaque. (**B**)—a large glomerulus is represented (arrow). (**C**–**E**)—separate frames from Appendix A, moving a large pendant. (**F**,**G**)—fluorescent and bright-field images of one fragment of the hypha; plaques are marked with arrowheads. Scale bars are 5 µm.

**Figure 3 jof-08-01316-f003:**
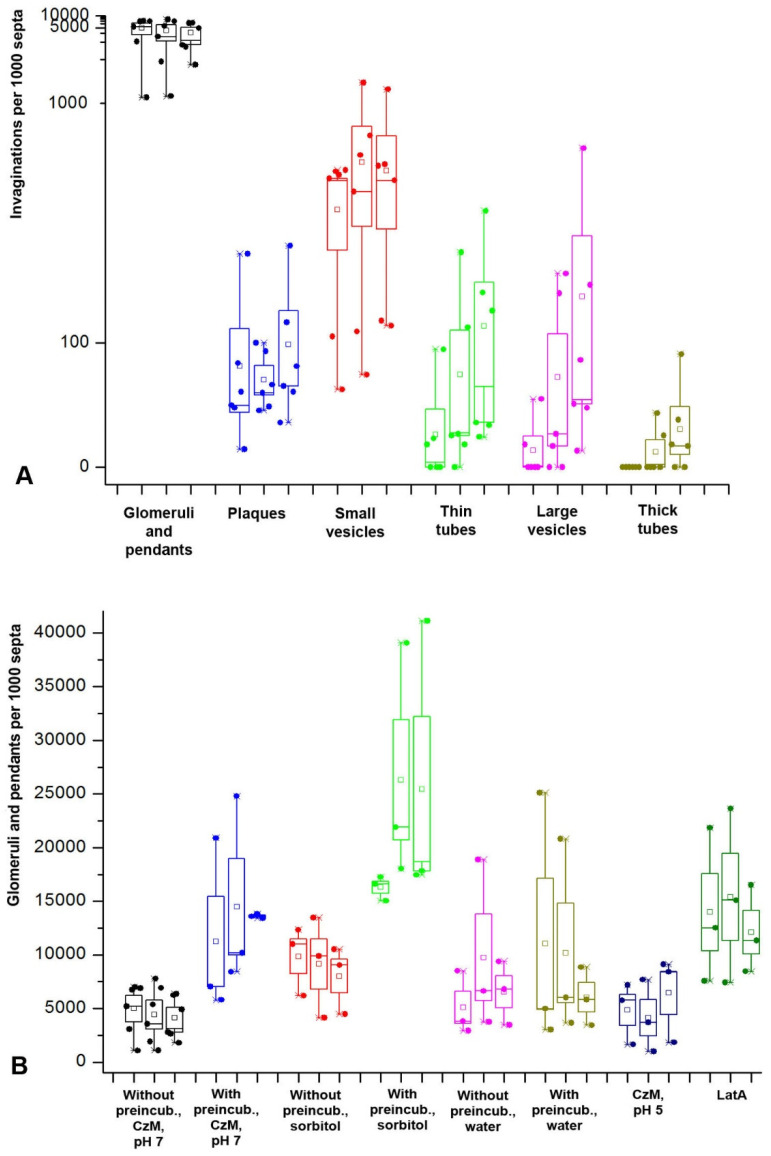
Diagrams showing the quantitative representation of different types of PM macroinvaginations in *S. hirsutum*. The diagram of each treatment variant or type of invagination consists of three subdiagrams—they correspond to the observation time intervals (5–15 min, 15–30 min, and 30–45 min). Each point on the diagram corresponds to the number of macroinvaginations observed in one biological experiment, calculated per 1000 septa. (**A**)—all types of macroinvaginations in the basic variant of treatment (without preincubation, CzM, pH 7). The Y-axis scale is non-linear. (**B**)—diagrams showing the number of glomeruli and pendants formed with different variants of sample treatment. Legend: small square—mean; box—50% standard deviation, the whisker–min/max.

**Figure 4 jof-08-01316-f004:**
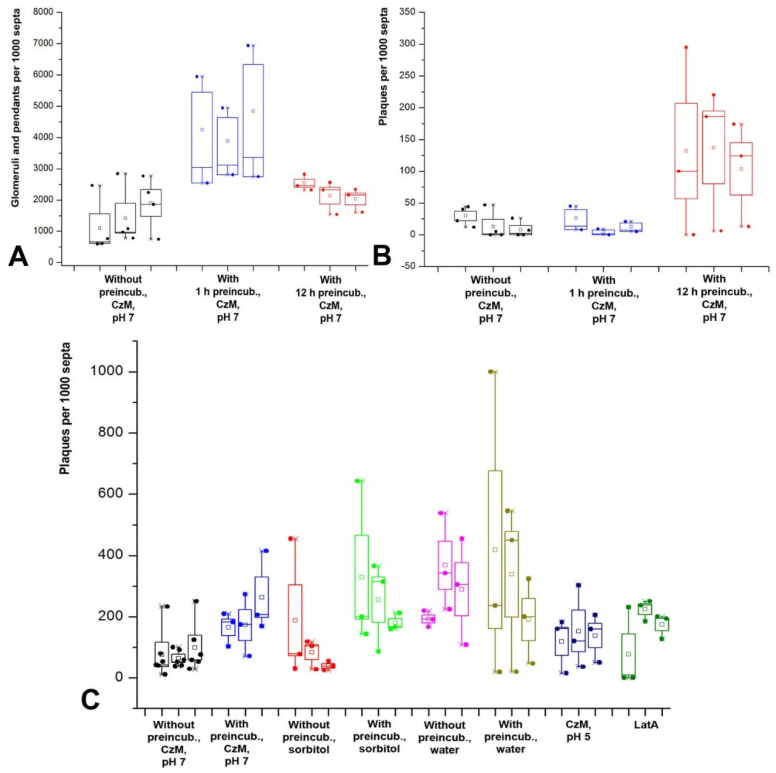
Diagrams showing the quantitative representation of plaques, glomeruli, and pendants in *S. hirsutum* and *O. olearius*. (**A**,**B**)—the diagrams reflect the influence of different variants of *O. olearius* mycelium preincubation on the formation of glomeruli and pendants (**A**) and plaques (**B**). (**C**)—formation of plaques in *S. hirsutum* with different types of sample treatment. Legend: small square—mean; box—50% standard deviation, the whisker–min/max.

**Figure 5 jof-08-01316-f005:**
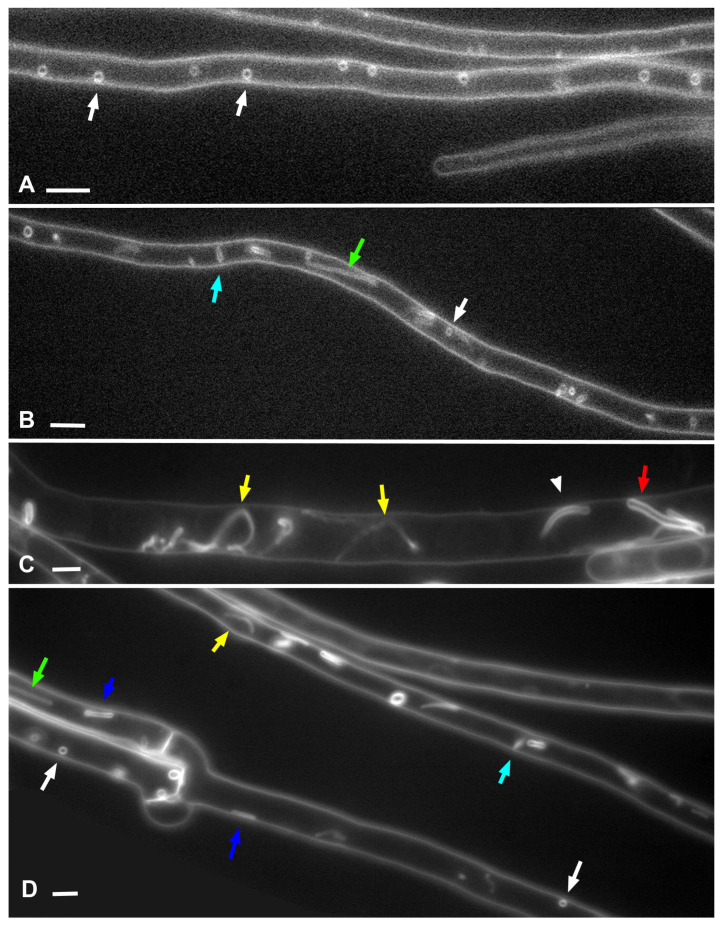
Small vesicles and thin tubes in the hyphae of *O. olearius*. Designations: white arrows—small vesicles, light blue arrows—standing thin tubes, blue arrows—longitudinal tubes pressed to the PM, green arrows—longitudinal straight tubes, yellow arrows—longitudinal curving tubes, red arrow—thick tube (>1 μm in diameter), arrowhead—tube of intermediate thickness (700–800 nm). (**A**,**B**)—the single focal planes. (**C**,**D**)—Z-stacks summation. Scale bars are 5 µm.

**Figure 6 jof-08-01316-f006:**
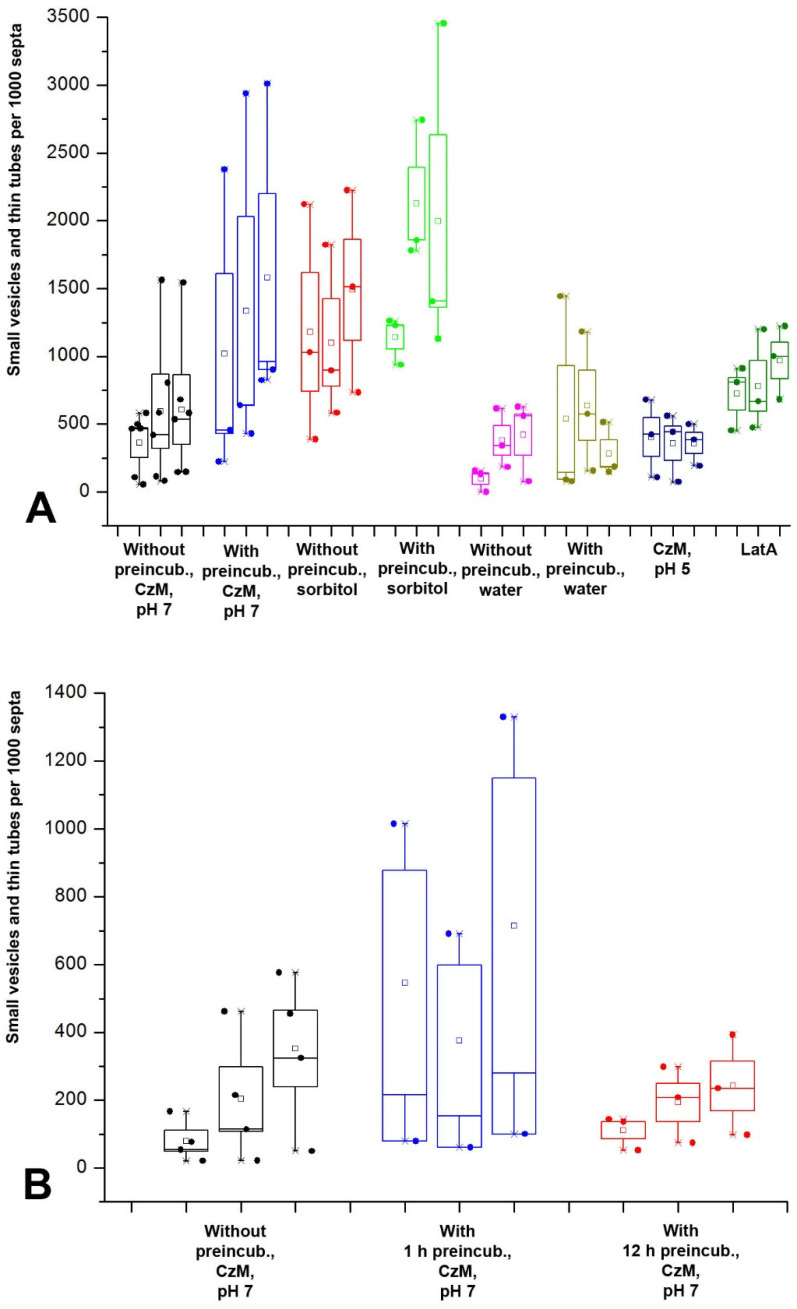
Diagrams showing the quantitative representation of small vesicles and thin tubes in *S. hirsutum* and *O. olearius*. (**A**)—influence of different variants of microscopic sample preparing on the formation of small vesicles and thin tubes in *S. hirsutum*. (**B**)—influence of the preincubation time on the formation of small vesicles and thin tubes in *O. olearius*. Legend: small square—mean; box—50% standard deviation, the whisker–min/max.

**Figure 7 jof-08-01316-f007:**
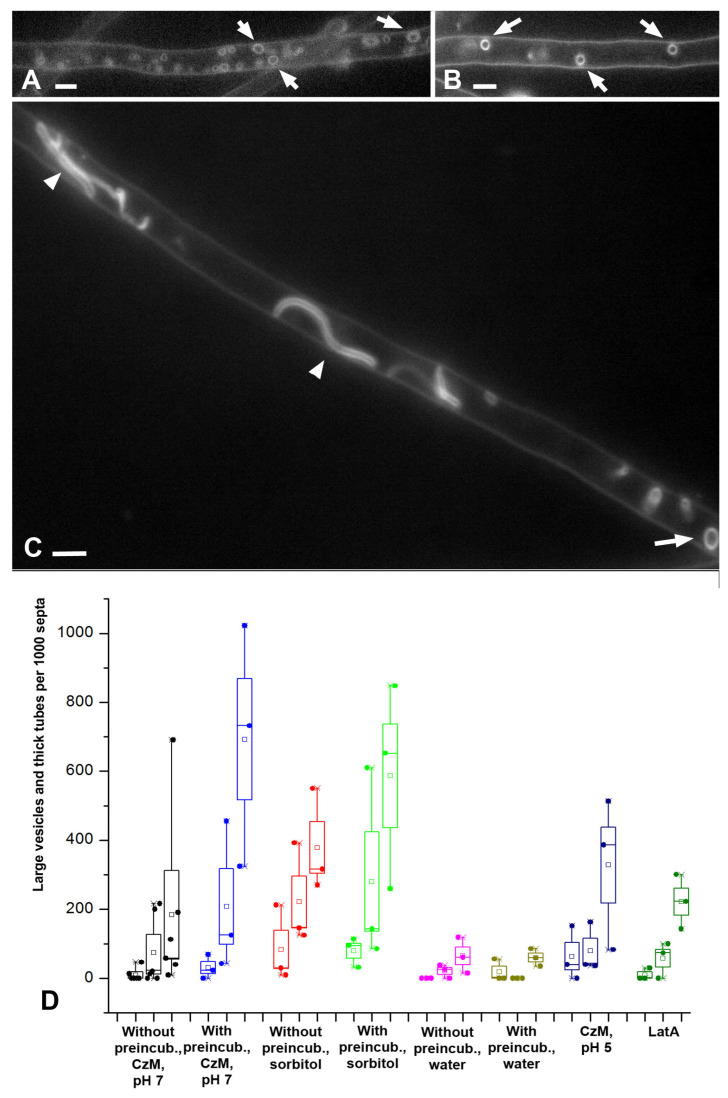
Large vesicles and thick tubes: fluorescent images and diagrams. Designations: white arrows—large vesicles, arrowheads—thick and intermediate tubes. (**A**,**B**)—hyphae fragments of *S. hirsutum* in single focal plane. (**C**)—Z-stacks summation of *O. olearius*. Scale bars are 5 µm. (**D**)—influence of different variants of microscopic sample preparing on the formation of large vesicles and thick tubes in *S. hirsutum.* Legend: small square—mean; box—50% standard deviation, the whisker–min/max.

**Figure 8 jof-08-01316-f008:**
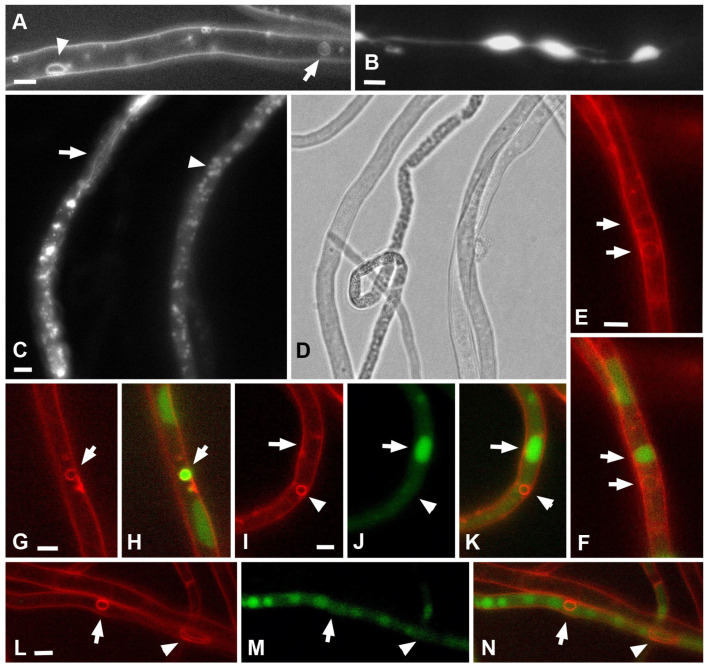
Vacuolar system and colocalization of PM macroinvaginations and vacuoles in *S. hirsutum*. (**A**)—AM4-64 labeling, arrowhead—large vesicle (or thick tube), arrow—VLC (vacuole-lysosome candidate). VLC differs from PM macroinvaginations by its often large size and weaker tonoplast signal. (**B**)—CFDA labeling, large oval vacuoles connected by tubular vacuoles. (**C**,**D**)—bright field and CFDA fluorescence images of small round vacuoles (arrowhead) and tubular vacuoles (arrow). (**E**,**F**)—labeling with AM4-64 (**E**); and co-labeling with AM4-64 and CFDA (**F**). Arrows indicate possible VLCs. It can be seen that only one of the two VLCs carries both labels. (**G**,**H**)—the only one of 119 large vesicles analyzed that carries both AM4-64 and CFDA labels (arrow). (**I**–**N**)—demonstration of the most common variant, when the vacuolar label and AM4-64 labeled large vesicles and thick tubes do not match. (**J**,**K**)—the arrow indicates a large central vacuole, and the arrowhead indicates a large vesicle. (**L**–**N**)—the arrow is directed to a large vesicle, and the arrowhead—to a thick tube. Scale bars are 5 µm.

**Figure 9 jof-08-01316-f009:**
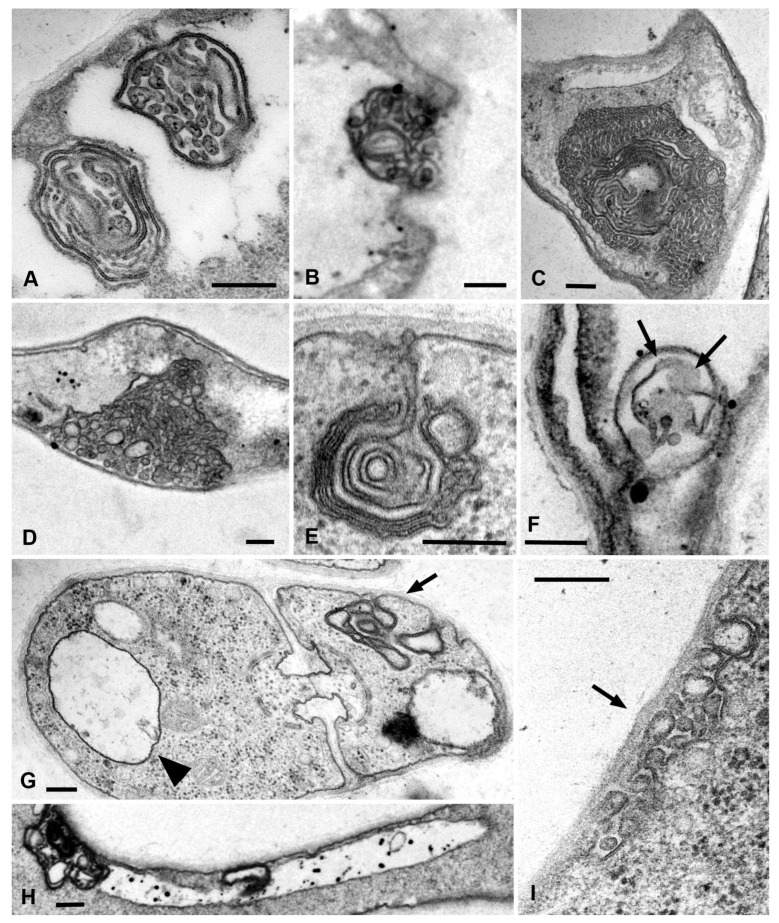
Transmission electron microscopic images of various types of lomasomes in xylotrophic basidiomycetes corresponding to various AM4-64 labeled PM macroinvaginations. (**A**)—classic lamellar (bottom) and vesicular (tubular, top) lomasomes corresponding to glomeruli (*P. luminescens*). (**B**)—lomasome (glomerulus) consisting of intertwined thin tubules (*L. edodes*). (**C**)—a complex lomasome consisting of ordered tubules and lamellae corresponds to a large glomerulus (*P. stipticus*). (**D**)—a complex vesicular (tubular) lomasome also corresponds to a large glomerulus (*P. luminescens*). (**E**)—a lamellar lomasome on a short stem corresponds to the pendant (*L. edodes*). (**F**)—a lamellar-tubular lomasome (*S. hirsutum*); it can be seen how the lamellae damaged during cutting turned their flat side into the section plane (arrows). (**G**)—a section through a dolipore septum (*L. edodes*); the arrow indicates the lamellar lomasome beginning to form; the arrowhead indicates the small vesicle. (**H**)—the thin longitudinal tube pressed against the PM at the ultrastructural level (*L. edodes*). (**I**)—a vesicular (tubular) lomasome corresponding to the small plaque (*P. luminescens*). Scale bars are 200 nm.

**Figure 10 jof-08-01316-f010:**
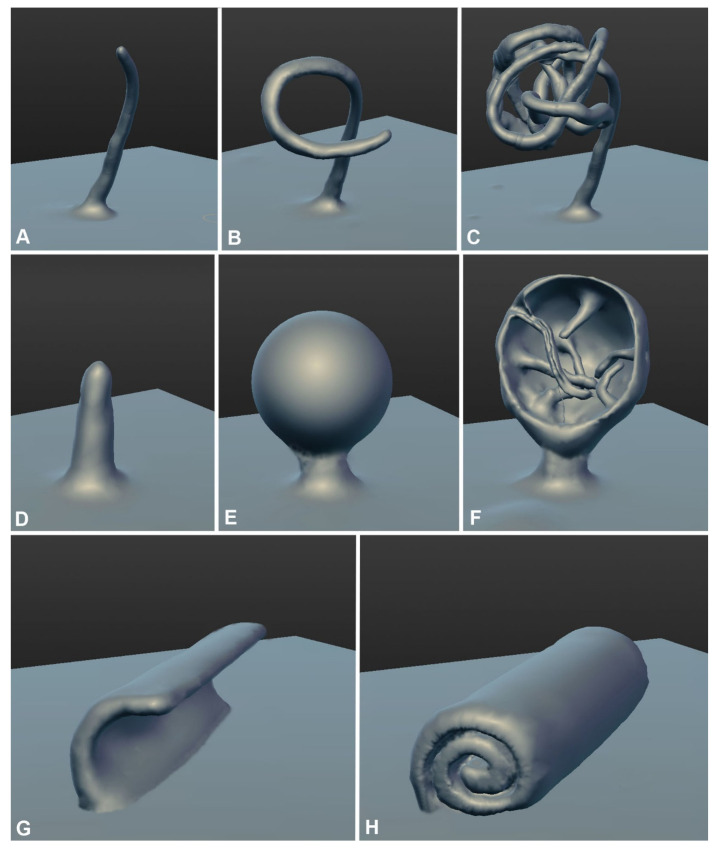
Three assumed mechanisms for the formation of glomeruli and pendants (same as lomasomes) in basidial xylotrophs. (**A**–**C**)—invagination of the PM in the form of a long filamentous tube, then the tube folds into a glomerulus (or a pendant if there is a free tubule-leg between the PM and the body of glomerulus). (**D**–**F**)—a thin invagination tube swells into a vesicle and then secondary tubular invaginations are formed from the vesicle membrane, filling the lumen of the maternal vesicle. (**F**)—the front of the maternal membrane is not drawn. (**G**,**H**)—a lamella, usually located across the cell, invaginates and then begins to roll up. The roller can be of different coil densities. Later, the lamella may also begin to swell and form secondary invaginations (not shown in the Figure).

## Data Availability

Not applicable.

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
