# Peer review of "Lomasomes and Other Fungal Plasma Membrane Macroinvaginations Have a Tubular and Lamellar Genesis"

_jof, 2022, doi:10.3390/jof8121316_

Round 1

Reviewer 1 Report

The paper made a comprehensive analysis on various macroinvagination types, suggests their biological functions and discusses some features of fungal endocytosis. The paper provides various video and pictures to verify the observations. The results are very interesting and helpful to make better understanding the complex fungal physiology. The paper also presents a new intracellular tubullar system in wood-decaying fungi.  Overall, the research was well designed and carried out. The presentation is clear. 

Author Response

We express our deep gratitude to the Reviewer for taking the time to analyze our manuscript and for the positive review.

Reviewer 2 Report

In this article, Mazheika et al use fluorescence and electron microscopy to understand the nature of a series of membranous invaginations and structures in xylotrophic basidyomicetes. Several of these structures had not been described before. In this sense, this work is a significant advance of knowledge in this field.

Also, I want to congratulate them on the high quality of the images and videos.

However, I have several concerns regarding the work, and the manuscript.

At experimental level:

1. If they really want to make sure that these structures do not correspond to endosomes/vacuoles, they must do co-staining with more specific dyes: quinacrine (green) or CMAC (blue).

2. Lomasomes are supposed to result in membranous structures embedded into the cell wall. They should do membrane and cell wall co-staining (for cell wall, Aniline blue, Calcofluor or Congo red can be used) to understand the relationship between these structures. Also, In encourage the authors to look carefully at the EM microscopy images to see if they find some of these structures (membrane portions trapped into the cell wall). In the micrographs shown in the article I have not seen membrane material trapped into cell wall material.

At conceptual level:

1. The fact that in hypotonic conditions the number of these structures increases can be related to changes in membrane tension, this should be discussed. Also, it is known that after hypotonic shock vacuoles fuse to generate larger vacuoles. Did they see a correlation between both processes?

2. Throughout the work (and in other works about lomasomes) it is assumed that these structures are related to endocitosis, but there is no proof about it. I have two important concerns. First, if the authors want to state this, they must localize some endocytosis marker (such as AP-2, or even actin) and show that the membranous invaginations they are describirng originate in the places where these markers localize. More importantly, LatA (which inhibits endocytosis) should inhibit the formation of these structures, and they do not see this effect with the actin-depolymerizing drug.

3. Membrane blebs have been described in cultured animal cells when they contact the plate surface; they form because of retraction of the cytoskeleton and are transient. Can any of the structures they see in this work be related to this phenomenon?

4. Membrane blebs and bubbles, which sometimes produce membrane material embedded into cell wall material have been described in Neurospora crassa, Saccharomyces cerevisiae and Schizosaccharomyces pombe mutants defective in Prm1 (the three organisms), or in Fig1 (S. cerevisiae)/Dni1 (S. pombe) proteins. These are proteins with an important role in membrane fusion. They should discuss the possible relationship between the structures they observe and membrane fusión events.

In summary, I think this is a nice descriptive work that must be improved with additional labeling experiments. However, much more detailed work is required to reach any functional conclusion. They must shorten considerably all the functional discussions from the article, because it is not supported by any result.

Minor:

Pge 1, line 31: change "eizosomes" by "eisosomes"

Round 2

Reviewer 2 Report

I am mostly satified with the answer given to my comments by the authors. I understand that the structures/proccesses they are describing ar quite diffferent from those known in ascomycetous yeasts and, therefore, cannot be interpreted in the same way. I thank their explanations